# Embryonic liver developmental trajectory revealed by single-cell RNA sequencing in the Foxa2$^{eGFP}$ mouse

Tianhao Mu [1,2,3,4,14], Liqin Xu [5,6,7,14], Yu Zhong [5,6,8,14], Xinyu Liu[4,9,14], Zhikun Zhao[5,6], Chaoben Huang[3], Xiaofeng Lan[3], Chengchen Lufei[4,9], Yi Zhou[4,9], Yixun Su [1,3], Luang Xu[9], Miaomiao Jiang[5,6], Hongpo Zhou[5,6], Xinxin Lin[5,6], Liang Wu[5,6], Siqi Peng[5,6], Shiping Liu [5,6], Susanne Brix [7], Michael Dean [10], Norris R. Dunn [11], Kenneth S. Zaret [12], Xin-Yuan Fu [1,2,3,4,9,13✉] & Yong Hou [5,6✉]

The liver and gallbladder are among the most important internal organs derived from the endoderm, yet the development of the liver and gallbladder in the early embryonic stages is not fully understood. Using a transgenic Foxa2$^{eGFP}$ reporter mouse line, we performed single-cell full-length mRNA sequencing on endodermal and hepatic cells isolated from ten embryonic stages, ranging from E7.5 to E15.5. We identified the embryonic liver developmental trajectory from gut endoderm to hepatoblasts and characterized the transcriptome of the hepatic lineage. More importantly, we identified liver primordium as the nascent hepatic progenitors with both gut and liver features and documented dynamic gene expression during the epithelial-hepatic transition (EHT) at the stage of liver specification during E9.5–11.5. We found six groups of genes switched on or off in the EHT process, including diverse transcriptional regulators that had not been previously known to be expressed during EHT. Moreover, we identified and revealed transcriptional profiling of gallbladder primordium at E9.5. The present data provides a high-resolution resource and critical insights for understanding the liver and gallbladder development.

[1] Department of Biochemistry, YLL School of Medicine, National University of Singapore, Singapore 119615, Singapore. [2] Laboratory of Human Diseases and Immunotherapies, West China Hospital, Sichuan University, 610041 Chengdu, China. [3] Department of Biology, Southern University of Science and Technology, 518055 Shenzhen, China. [4] GenEros Biopharma, 310018 Hangzhou, China. [5] BGI-Shenzhen, 518033 Shenzhen, China. [6] China National GeneBank, BGI-Shenzhen, 518120 Shenzhen, China. [7] Department of Biotechnology and Biomedicine, Technical University of Denmark, Soltofts Plads, 2800 Kongens Lyngby, Denmark. [8] School of Biology and Biological Engineering, South China University of Technology, 510006 Guangzhou, China. [9] Cancer Science Institute of Singapore, YLL School of Medicine, National University of Singapore, Singapore 117599, Singapore. [10] Laboratory of Translational Genomics, Division of Cancer Epidemiology & Genetics, National Cancer Institute, Gaithersburg, MD, USA. [11] Endodermal Development and Differentiation Laboratory, Institute of Medical Biology, Agency for Science, Technology and Research (A*STAR), Singapore 138672, Singapore. [12] Institute for Regenerative Medicine, University of Pennsylvania, Perelman School of Medicine, Smilow Center for Translation Research, Philadelphia, PA 19104, USA. [13] State Key Laboratory of Biotherapy, West China Hospital, Sichuan University, 610041 Chengdu, China. [14]These authors contributed equally: Tianhao Mu, Liqin Xu, Yu Zhong, Xinyu Liu. ✉email: bchfxy@nus.edu.sg; houyong@genomics.cn

The liver is the largest internal organ and provides many essential metabolic, exocrine, and endocrine functions, including the production of bile, the metabolism of dietary compounds, detoxification, regulation of glucose levels, and control of blood homeostasis through secretion of clotting factors and serum proteins such as albumin (Alb)[1]. After gastrulation, the foregut endoderm is derived from the primitive streak (PS) at mouse embryonic day 7.5 of gestation (E7.5)[2]. The liver is derived from the foregut endoderm, and the hepatic marker Alb is first detected in the nascent hepatic endoderm within the 7–8 somite stage at E8.5[3,4]. Foxa2 has been considered as an endoderm marker at E6.5 and is expressed in all the differentiated endoderm-derived organs, including the liver[5]. FOXA2 acts as a "pioneer" factor in liver development and serves to de-compact chromatin at its target sites[6]. Disruption of FOX factors (Foxa2, Foxh1), GATA factors, Sox17, Mixl1, or SMAD signaling all lead to defects in the gut tube and liver morphogenesis[7–12]. During liver specification, a portion of the gut tube cells receives fibroblast growth factor (FGF) signals from the developing heart[3] and bone morphogenetic protein (BMP) from the septum transversum mesenchyme (STM)[13]. This leads to the differentiation of the hepatoblast, which constitutes the liver primordium or liver bud at E10.5[14,15]. Several transcription factors (TFs) have shown to be essential for liver specification, including Tbx3, Hnf4a, and Prox1[16–18]. Primarily, the program of hepatogenesis has been studied by conventional immunohistochemistry and analysis of tissue explants; however, a complete pattern of transcriptional dynamics during liver specification remains to be unveiled due to the difficulties of isolation of pure nascent hepatic progenitors.

The liver primordium, primitive gallbladder, and primitive pancreas arise from the foregut endoderm at almost the same time at E9.5[19–21]. The PDX1+ and SOX17+ pancreatobiliary progenitor cells segregate into a PDX1+/SOX17− ventral pancreas and a SOX17+/PDX1− biliary primordium[22]. In another study, Lgr4 has been shown to be significant for gallbladder development since Lgr4 depletion affects the elongation of the gallbladder, but has no effect on the liver bud and ventral pancreas[23]. Apart from such studies, the molecular features and drivers of gallbladder development are unexplored.

Recently, two studies characterized the landscape of the gut endoderm, at E3.5–E8.75 and E6.5–E8.5, respectively, by using single-cell RNA sequencing[24,25]. Two other studies focused on liver differentiation from E10.5 or 11.5 onwards and discerned the split between the hepatocyte and cholangiocyte lineages[26,27]. However, liver specification, the key process that liver primordium differentiated from the gut tube at E9.5, has not been described on a single-cell level. In the mouse embryo single-cell atlas study, the organogenesis landscape from E9.5 to E13.5 was characterized using sci-RNA-seq3[28]. However, quantities of transcriptional information might be lost, considering the low-detected gene number (519 genes per cell on average). Thus, a high-quality single-cell RNA-seq dataset generated with high-sensitive methods is demanded to improve the understanding of liver development.

In this study, we constructed a transgenic Foxa2eGFP reporter mouse line to trace the endodermal and hepatic cells in the early stages of development. By applying single-cell full-length mRNA sequencing of 1966 single cells from endodermal and hepatic regions from E7.5 to E15.5, we have identified the endoderm and hepatic lineages and characterized the key networks and transcription factors responsible for endodermal morphogenesis and liver development. We also identified the gallbladder primordium at E9.5 and found it could be distinguished transcriptionally from liver primordium. Our data provide a resource for further research into endodermal differentiation and liver development, which could potentially lead to therapeutically useful tissue for liver transplantation.

## Results

### Foxa2eGFP tracing of endoderm and hepatic cells and scRNA sequencing.
To access purified endodermal and hepatic-related cells, we generated a transgenic Foxa2eGFP reporter mouse line (Fig. 1a and Supplementary Fig. 1). In this mouse model, the eGFP (enhanced green fluorescent protein) gene was linked to the third exon of Foxa2 (Fig. 1a). Homozygous transgenic mice develop normally and did not show an abnormal phenotype. As expected for the endogenous Foxa2 gene[29–31], we found eGFP to be expressed in the mouse embryo labeling the endoderm, neural system, and endoderm-derived organs, including the liver (Fig. 1b, c). The fluorescence in the liver was impaired due to the perfusion of hematopoietic cells from E11.5, but the fluorescence was evident upon liver dissection. Immunofluorescence assay showed that hepatoblasts expressed FOXA2 and DLK1 at E12.5 simultaneously in Foxa2eGFP mice (Fig. 1d and Supplementary Fig. 1e). We dissected the distal half part of the whole embryo at E7.5 and the foregut endoderm at E8.5, the hepatic region from E9.5, E10.0, and E10.5, and the whole liver from E11.5, E12.5, E13.5, E14.5, and E15.5, including two replicates at E12.5, E13.5, and E14.5 (Fig. 1b). At E11.5, the liver was precisely dissected, excluding the pancreas, lung, and stomach (Fig. 1c).

To characterize endoderm and liver development, we performed single-cell full-length mRNA-Seq experiments on the Foxa2eGFP+ cells isolated from E7.5 to E15.5 (Fig. 1e). The tissues were dissociated into a single-cell suspension, and Foxa2eGFP+ cells were sorted into 96-well plates with one cell in each well using the BD FACSAria™ III cell sorter. Doublets and multiplets were excluded by analysis of side scatter (SSC) and forward scatter (FSC) (Supplementary Fig. 2). The amplified cDNA was assessed by agarose gel and qPCR of Afp, a hepatic marker gene, before library generation (Supplementary Fig. 3). In total, 1246 individual cells were collected and the mRNA amplified following the SMART-seq2 protocol. In addition, we generated libraries from 720 cells from E11.5, E12.5, and E13.5 by MIRALCS (microwell full-length mRNA amplification and library construction system)[32]. All together, the transcriptomes of 1966 individual cells, as well as bulk control samples from 10 embryo stages, were sequenced.

The 1246 SMART-seq2 cells were used to identify cell populations during liver development. After filtering unqualified reads, gene expression levels were characterized by reads per kilobase per million mapped reads (RPKM) with RPKM > 1 as the threshold (Supplementary Fig. 4). To obtain high-quality cells for subsequent analysis, we removed cells with fewer than 6000 expressed genes (RPKM > 1) since most of those cells express low levels of Gapdh (Supplementary Fig. 5b). We obtained 922 cells with an average of 9378 genes with RPKM > 1 and an average of 9.5 million mapped sequencing reads (Supplementary Fig. 5a, c and Supplementary Data #1). Technical noise was assessed by bulk sample sequencing, experimental replicates, and sequencing batch effect analysis (Supplementary Fig. 6a–d), confirming that the final dataset was of high quality and reliable. As eGFP and Foxa2 were co-expressed in our mouse model, a high correlation was detected (Pearson $r = 0.95$) between these two genes (Supplementary Fig. 6e).

### Cell clustering and identification of cell types.
We clustered and visualized the high-quality 922 cells from 10 embryonic stages range from E7.5 to E15.5 using t-distributed stochastic neighbor embedding (t-SNE)[33] (Supplementary Fig. 7a). Eight major cell populations were defined based on the expression of marker genes (Fig. 1f, g). The primitive streak cells were identified at E7.5 based on the expression of genes related to gastrulation, including Pou5f1, Mixl1, Lefty2, Cer1, Cyp26a1, Lhx1, Fgf5, Hesx1, and

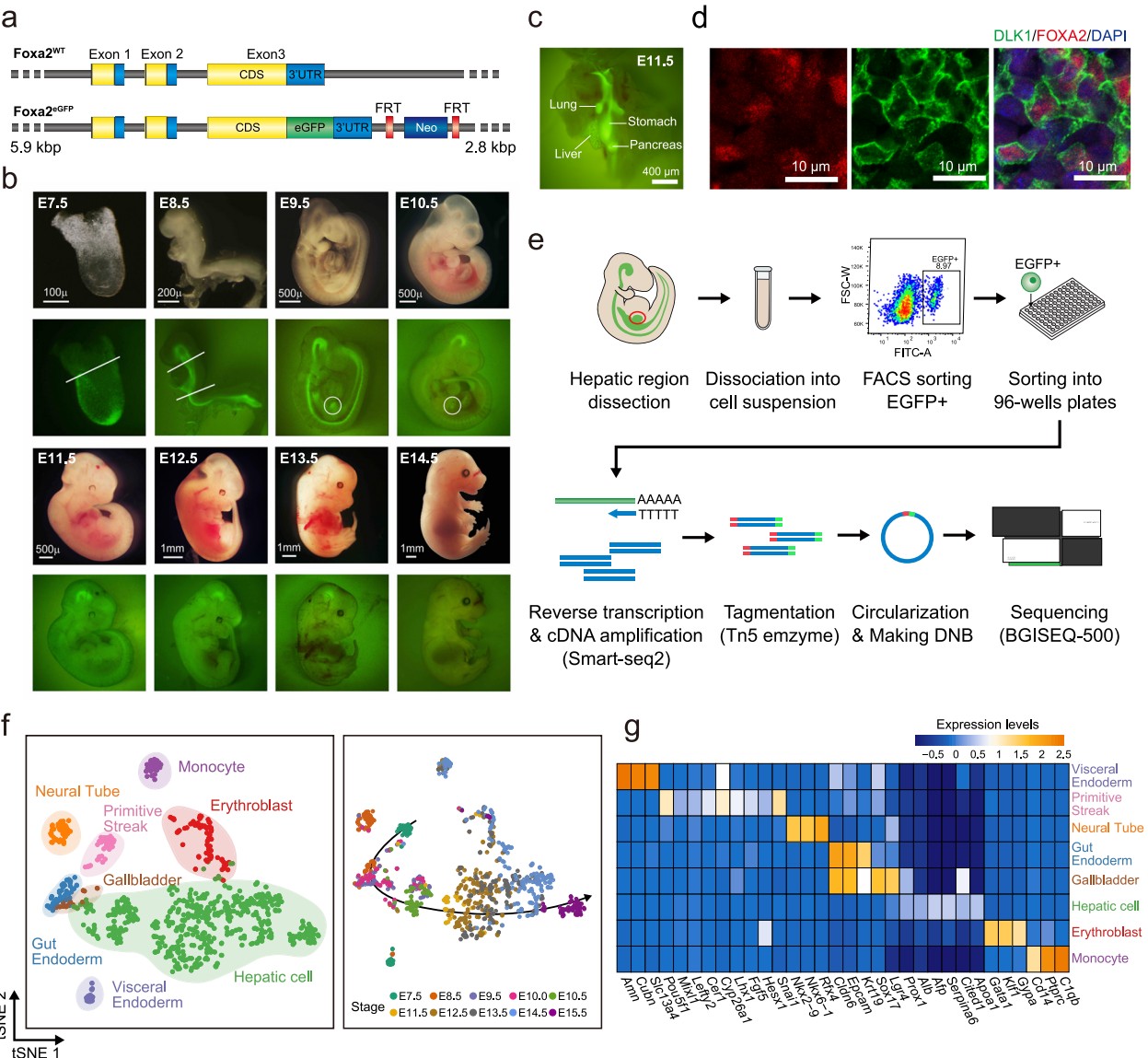

**Fig. 1 Single-cell full-length RNA-seq to analyze liver development during E7.5–E15.5 by using a Foxa2 eGFP mouse model. a** The vector design for Foxa2eGFP mouse. *eGFP* is linked to the third exon of *Foxa2*. CDS coding sequence, 3′UTR 3′ untranslated region, FRT flippase recognition target, Neo Neomycin. **b** eGFP-labeled mouse embryos from E7.5 to E14.5. The endoderm, neural system, and endoderm-derived organs, including liver expressed eGFP. The general dissection strategies are shown (white lines or circles). Scale bars, 100 μm for E7.5; 200 μm for E8.5; 500 μm for E9.5–E11.5; 1 mm for E12.5–E14.5. **c** Precise dissection for the endodermal organs at E11.5. The liver, lung, stomach, and pancreas are shown. Scale bars, 400 μm. **d** Immunofluorescence analysis of paraffin-sectioned mouse embryo at E14.5, showing co-expression of FOXA2 (red), DLK1 (green), and DAPI (blue) in hepatoblasts. Scale bar, 10 μm. **e** The workflow of single-cell full-length RNA sequencing of Foxa2eGFP+ cells. **f** t-SNE visualization of embryonic cells from E7.5 to E15.5, colored by cell populations and embryonic stages. The left panel shows the cell identity of each cell cluster, and the right panel shows the embryonic stage of every single cell. The developmental route (shown by the arrow in the right panel) by t-SNE agrees with the embryonic stages of single cells. **g** Expression of selected marker genes for the identified cell clusters. The averaged gene expression level of marker genes are shown by different colors.

*Snail1*. The primitive streak cells were validated by applying these cells to the iTranscriptome database[34] (Supplementary Fig. 7b). Visceral endoderm cells with the expression of *Amn, Cubn*, and *Slc13a4* were also found at E7.5. A group of gut endoderm cells which express *Cldn6* and *Epcam* was mainly found at E8.5, E9.5, and E10.0. The neural tube (*Nkx2-9*), which is spatially close to the gut endoderm, was mainly found at E8.5, with a small number of cells detected at E9.5 and E10.0. The hepatic cell group that expresses *Prox1* and *Alb* was found at the stages range from E9.5 to E15.5. Three pancreas-like cells that expressed *Pdx1* but not *Sox17* were identified and excluded from further analysis due to the limited cell number. Moreover, we identified a group of

gallbladder primordium (GBP) cells mainly at E9.5 that expressed *Sox17* and *Lgr4*, but not *Pdx1*. Erythroblast cells expressing *Gata1* and monocytes expressing *Ptprc* (CD45) were identified at the stages from E11.5 to E15.5.

**Transcriptional profiling of the gut endoderm.** To investigate the transcriptional profiling of the gut endoderm before liver specification, we re-clustered *Foxa2+eGFP+* cells in E7.5 and E8.5 (Supplementary Fig. 7a) and identified the gut endoderm cells from E8.5 (Fig. 2a). Previous studies have revealed that gut endoderm comprises cells of both visceral endoderm and definitive endoderm decendants[24]. We observed a part of the gut

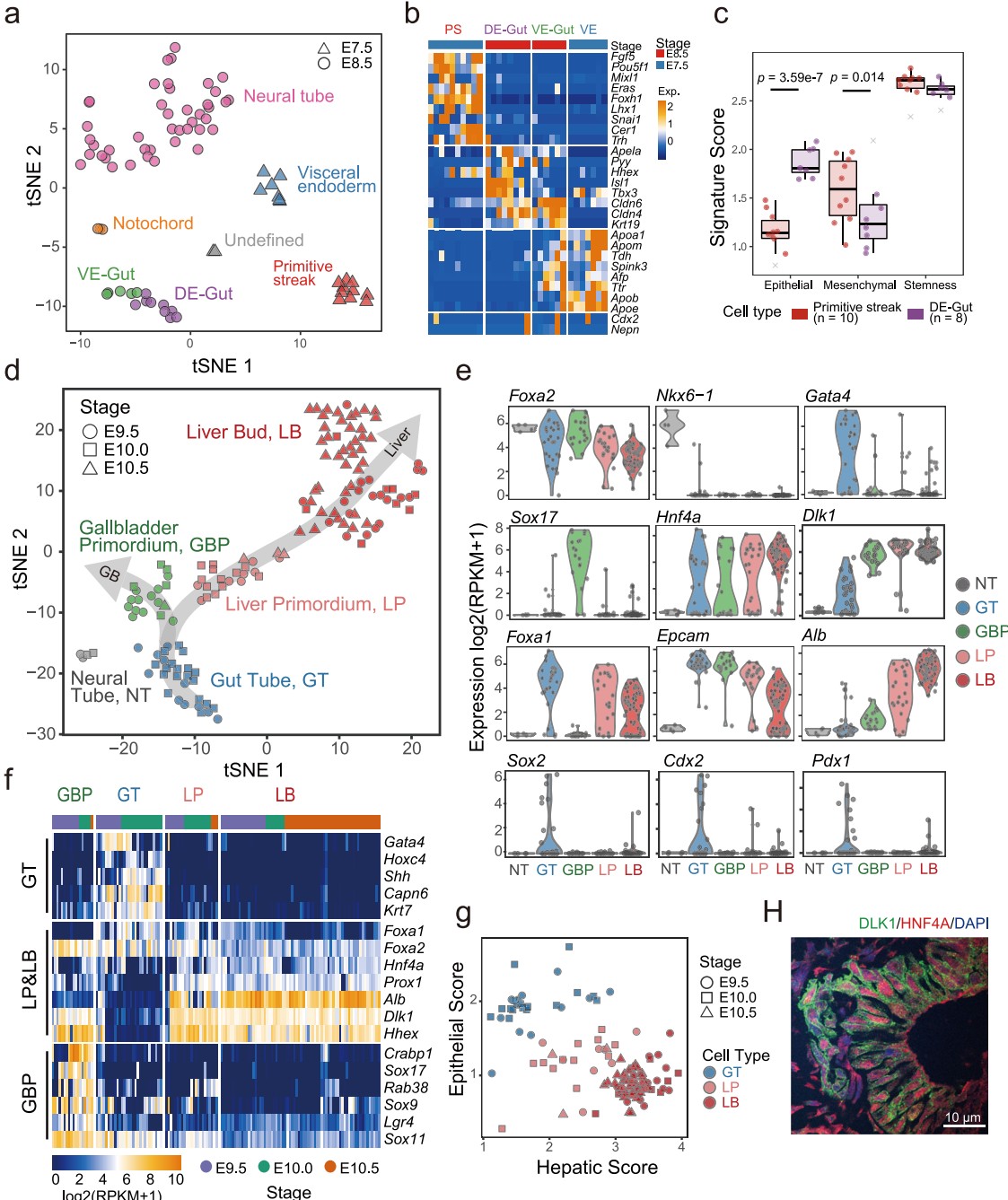

**Fig. 2 Identification of nascent hepatoblasts and gallbladder primordium at E9.5. a** t-SNE visualization of single cells at E7.5 and E8.5. DE-Gut definitive endoderm-derived gut, VE-Gut visceral endoderm-derived gut. **b** Specific markers for PS, VE, DE-Gut, and VE-Gut in panel **a**. PS primitive streak, VE visceral endoderm. **c** Box plots of the epithelial feature, mesenchymal feature, and stemness feature of the PS and DE-Gut. Crosses in gray represent outliers that were not included in the statistical test (*t* test). **d** t-SNE visualization of single cells from E9.5, E10.0, and E10.5. GT Gut tube, LP liver primordium, LB liver bud, GBP gallbladder primordium, NT neural tube. **e** Violin plots of the gene expression of specific markers in the cell populations at E9.5, E10.0, and E10.5. **f** The heatmap illustrating the normalized expression of the differentially expressed genes of the gut tube, liver primordium, liver bud, and gallbladder primordium. **g** The dot plot illustrating the epithelial feature and hepatic feature of GT, LP, and LB cells. **h** Immunofluorescence showing the co-expression of DLK1 (green) and HNF4A (red) in the hepatic region of E9.5 embryo. Nuclei were stained with DAPI. Scale bar, 10 μm.

endoderm cells expressed high levels of *Apela*, *Hhex*, *Isl1*, and *Tbx3* which were characteristics of the definitive endoderm (DE-derived gut endoderm, DE-Gut), while the others expressed high levels of *Afp*, *Ttr*, and apolipoprotein genes which together were characteristics of the visceral endoderm (VE-derived gut endoderm, VE-Gut) (Fig. 2a, b, Supplementary Data #2). Moreover, we found the DE-Gut highly expressed foregut endoderm marker

(*Pyy*), but not midgut (*Nepn*) or hindgut (*Cdx2*) markers except for one cell, which was consistent with our tissue dissection (Fig. 2b).

To characterize gene regulation during gut morphogenesis, we compared the gene expression of primitive streak and DE-Gut (Supplementary Fig. 7c and Supplementary Data #3). We found the expression of transcription factor genes (*Pou5f1*, *T*, *Eomes*,

*Hesx1*, and *Zic3*) related to cell pluripotency was decreased in DE-Gut, compared with the primitive streak (Supplementary Fig. 7c). By analyzing the expression of feature genes (Supplementary Data #4), we found the epithelial features increased, but mesenchymal features decreased in DE-Gut during the gut endoderm morphogenesis (Fig. 2c). Differentially expressed genes were analyzed by ingenuity pathway analysis (IPA), indicating that the Wnt/β-catenin signaling pathway was repressed by *Sox11*, *Rxrb*, and *Tgfb2* in the DE-Gut, while BMP signaling was activated by downregulation of the BMP suppressor *Fst* and *Chrd* (Supplementary Fig. 7d, e). These results are consistent with previous studies, which found that Wnt signaling initially suppressed mammalian liver induction[35,36], while BMP signaling helped induce it[13].

**Identification of nascent hepatoblasts and gallbladder primordium at E9.5.** To characterize the heterogeneity of hepatic cells and identify nascent hepatoblasts during liver specification, we re-clustered and visualized all the *Foxa2+eGFP+* cells from E9.5, E10.0, and E10.5 by t-SNE (Fig. 2d and Supplementary Fig. 7a). Undifferentiated gut tube (GT) with epithelial features were identified to express high levels of *Epcam*, *Gata4*, and *Shh* (Fig. 2e, f). Differentiated hepatic cells were identified based on the expression of well-known marker genes, such as *Alb, Hnf4a, Hhex, Prox1*, and *Dlk1*. *Epcam* decreased during the differentiation of hepatic cells, while *Shh* and *Gata4* were almost completely silenced during liver specification, consistent with previous reports[37,38] (Fig. 2e, f). We found that the hepatic cells were clustered into two groups based on the whole-transcriptome difference, and we defined them as liver primordium (LP) and liver bud (LB) by two criteria (Fig. 2d). First, most of the LP cells were found in E9.5 and E10.0, while the LB cells were mainly found at a later stage E10.5. Second, LP cells have higher expression of the epithelial marker *Epcam* but lower expression of the hepatic marker *Alb* compared with LB (Fig. 2e). For confirmation, we quantified the hepatic features of these single cells with a hepatic score using the expression of a set of genes related to hepatic functions (Supplementary Data #4). By analyzing the hepatic and epithelial score of the gut tube, liver primordium, and liver bud, we found the epithelial score decreased while the hepatic score increased during liver specification. Liver primordium exhibited an intermediate state between the undifferentiated gut tube and differentiated liver bud (Fig. 2g). The transitional process, epithelial–hepatic transition (EHT), from the endoderm with epithelial characteristics to the liver bud with hepatic characteristics, was observed, and is remarkably consistent with cytological changes reported during this period[15]. During the EHT, LP presented both epithelial features and hepatic features, indicating that LP cells were the nascent hepatic cells that differentiated from the gut tube. DLK1 was reported as a surface marker of hepatoblasts and able to isolate hepatoblasts, but not reported at E9.5[39]. We found that LP expressed both *Dlk1* and *Hnf4a* at E9.5 (Fig. 2e), and this was validated by immunofluorescence (Fig. 2h). Moreover, DLK1 can be used to isolate nascent hepatoblasts (LP) by FACS during liver specification (Supplementary Fig. 8a).

In addition to the hepatic cells, we identified a group of gallbladder primordium (GBP) cells which expressed *Sox17* and *Lgr4*, but not *Pdx1* at E9.5 (Fig. 2d, e). Taken together with the gut tube and liver primordium cells, a two-direction developmental trajectory of the gut tube was identified (Fig. 2d). Interestingly, the gallbladder primordium cells express many hepatic genes, including *Alb* and *Dlk1*, but not as high as liver primordium (Fig. 2e). Moreover, *Foxa1* was expressed in the gut tube and liver primordium but negative in the gallbladder primordium, while the expression of *Foxa2* was positive in both liver primordium and gallbladder primordium, indicating that *Foxa1* was selectively suppressed during gallbladder development. By differentially expressed genes analysis, 411 genes were found to be upregulated in the gallbladder primordium compared with the gut tube from E9.5 to E10.5 (Supplementary Data #5). This 411 genes group included the *Sox* family genes *Sox9*, *Sox11*, and *Sox17* (Fig. 2f). *Sox9* has been reported to be related to cholangiocyte differentiation[40]. In addition, *Crabp1, Rab38, Flt1, Slco5a1, Ptpn5, Vstm2b, Ntrk2, Ets1*, and *Ypel4* were identified as potential markers in the gallbladder primordium, while barely expressed in the gut tube and hepatic cells (Fig. 2f and Supplementary Fig. 8b). By contrast with gut tube and liver primordium, *Junb, Hpx, Mt2, Lrrc3, Dkk3, Apob, Acss1*, and *Dhrs3* were barely detected in the gallbladder (Supplementary Fig. 8b).

**Major gene expression dynamics during the epithelial–hepatic transition (EHT).** To characterize the epithelial–hepatic transition process, we analyzed the differentially expressed genes between the gut tube (GT), liver primordium (LP), liver bud (LB), and hepatic cells from E11.5 (E11.5 Hep). The heatmap of differentially expressed genes demonstrated a programmed change of gene expression from the gut tube to the hepatoblast (Fig. 3a). This set of 202 genes could be clustered into six gene groups: L1, L2, L3, G1, G2, and G3 (representing the following gene groups: Liver 1, Liver 2, Liver 3, Gut tube 1, Gut tube 2, Gut tube 3, respectively), based on temporal order and biological functions (Fig. 3a and Supplementary Data #6). These gene groups were dynamically regulated by the developmental axis GT–LP–LB–Liver (Fig. 3b). During liver specification, L1 genes were first switched on in liver primordium, followed by L2 in the liver bud, and L3 in E11.5 liver. Meanwhile, G1, G2, and G3 genes were downregulated or switched off in the liver primordium, liver bud, and liver in E11.5, respectively. The L1 genes were enriched in blood coagulation and hemostasis (genes including *F10, F12, Fga, Serpina6*, and *Serpind1*) and lipid metabolic processes (*Apoc2* and *C3*) (Fig. 3c). L2 genes were related to oxidation–reduction (such as *Cyp2d10, Sord*, and *Hsd17b2*) and triglyceride catabolism (*Aadac* and *Apoh*). The L3 genes were involved in the glucose metabolic process and fatty acid oxidation (*Pdk4, Cpt1a*, and *Slc27a5*). Meanwhile, in the G1, G2, and G3 gene groups that decreased during liver development, we have identified many epithelial feature genes (including collagen, claudin, and laminin). The *Grhl2* gene was first downregulated in the liver primordium, the *Kit, Krt19*, and *Col2a1* were then downregulated in the liver bud, and finally, the *Epcam* and *Cldn6* almost disappeared in the liver of E11.5. In summary, extensive change in gene expression patterns was a dominant feature during the EHT.

We validated these 202 EHT genes with the other two single-cell datasets[27,28]. Yang et al.[27] described the hepatic cell development during E10.5–E17.5 by full-length scRNA-Seq. We found the similar gene expression pattern of the six gene groups in the dataset, except for their lacking of cells from E9.5 (Supplementary Fig. 9a). The other study identified hepatic cells from E9.5 to E11.5 using sci-RNA-seq3; however, it was found that only a very small number of EHT genes were detected in the dataset due to the low sensitivity of the 3′ end RNA-Seq method (Supplementary Fig. 9b).

**Transcription factors and RXR complex signaling dynamics during liver specification.** To identify the genes that trigger the hepatic fate, we focused on the 548 genes encoding

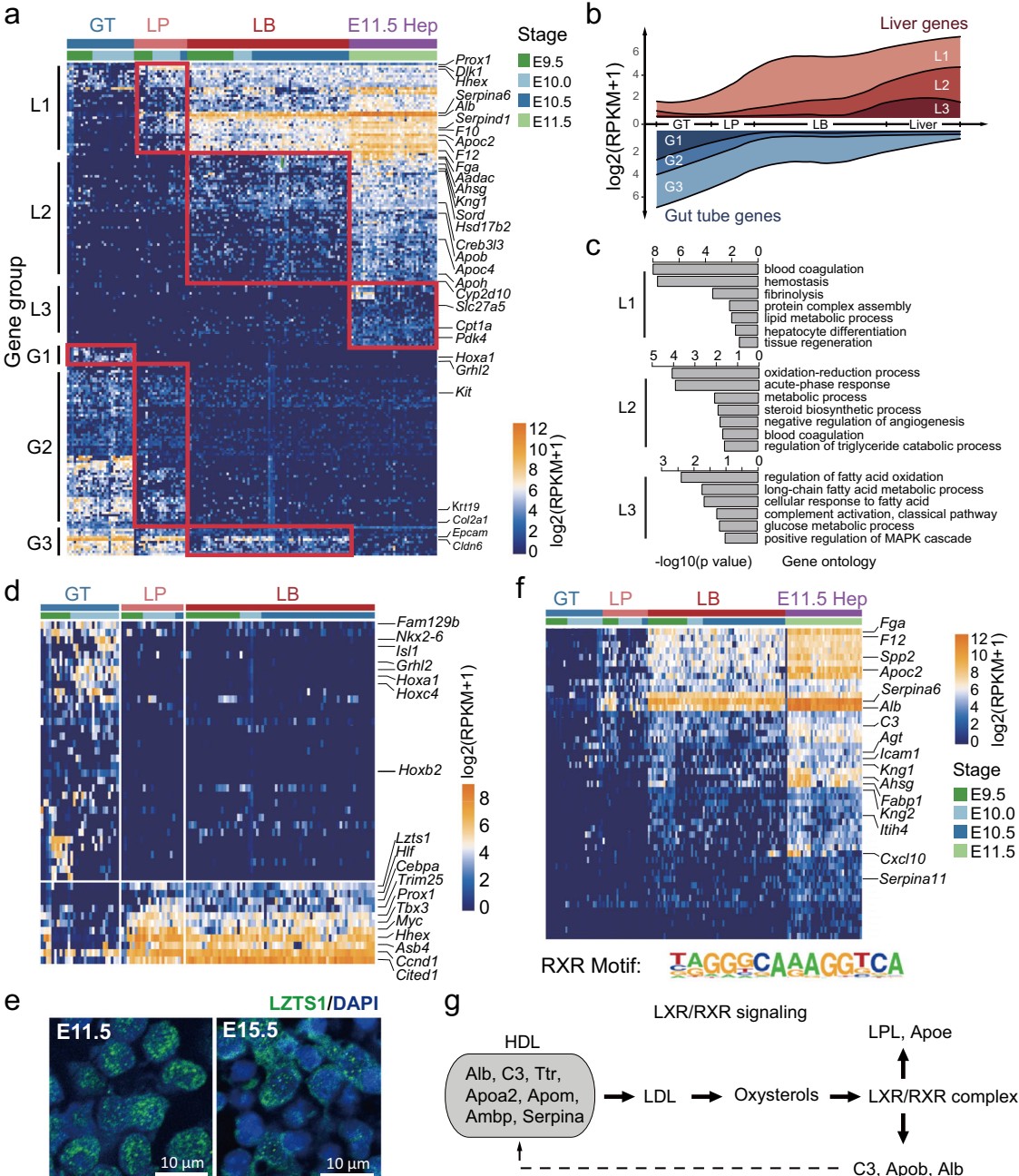

**Fig. 3 Dynamic gene expression of the epithelial–hepatic transition (EHT) during liver specification. a** Heatmap of differentially expressed genes of the gut tube, liver primordium, liver bud, and E11.5 hepatoblasts. Six groups of differentially expressed genes were identified to be switched on or off, including three liver gene groups: L1, L2, and L3, and three gut tube gene groups: G1, G2, and G3. GT gut tube, LP liver primordium, LB liver bud, E11.5 Hep hepatoblasts at E11.5. **b** The gene expression levels of six gene groups (L1, L2, L3, G1, G2, G3) were identified, followed by the development axis GT–LP–LB–Liver. **c** Gene ontology of the gene groups L1, L2, L3 were identified (*P* < 0.01). **d** Transcription factors that differentially expressed between the gut tube and liver primordium were identified, including some new transcription factors such as *Lzts1, Hlf, Trim25, Myc, Asb4, Ccnd1*, and *Cited1*. **e** Immunofluorescence showing the expression of LZTS1 in hepatoblasts at E11.5 and E15.5, respectively. Nuclei were stained with DAPI. Scale bar, 10 μm. **f** RXR motif was identified in the genes highly expressed in liver primordium. The 49 targets of RXR motif were shown. **g** The LXR/RXR signaling pathway was significantly upregulated in the liver primordium compared with the gut tube and formed a positive-feedback loop.

transcription factors, enzymes, cytokines, transporters, and kinases that were differentially expressed between the gut tube and liver primordium (Supplementary Data #7 and Supplementary Fig. 10a). In total, 49 TF genes were found to be activated in the liver primordium, including *Cebpa, Prox1, Tbx3*, and *Hhex* as expected. Moreover, we have identified several upregulated TF genes, including *Lzts1, Hlf, Trim25, Myc, Asb4, Ccnd1*, and *Cited1* (Fig. 3d). Mutation of the mouse

*Lzts1* gene has been reported to result in hepatocellular carcinoma[41]. We validated the expression of *Lzts1* by immunofluorescence and found it was highly expressed at E11.5 and E15.5 (Fig. 3e). We also identified TF genes downregulated in the liver primordium, including *Hoxa1, Hoxb2, Hoxc4, Grhl2, Isl1, Nkx2-6*, and *Fam129b*. Target genes (*Cdh1, Cldn4, Sema3c, Sema3b, Rfx2, Nrp2*) of *Grhl2* were also found to be downregulated (Supplementary Fig. 10b).

The gene networks and signaling pathways of the 548 differentially expressed genes between gut tube and liver diverticulum were enriched in "Cellular Development", "Cell Growth and Proliferation", "Connective Tissue Development and Function", "Embryonic Development", and "Organismal Development" (Supplementary Fig. 10c). More importantly, we found the liver X receptors/retinoid X receptors (LXR/RXR) pathway was significantly upregulated in the liver primordium compared with the gut tube, including *Alb, Ambp, ApoA1, ApoA2, ApoE, ApoF, ApoM, C3, Ttr, SerpinA1, SerpinF1, SerpinF2,* and other genes (Supplementary Fig. 10d).

To validate the role of the LXR/RXR pathway, we analyzed the promoters of the differentially expressed genes between the gut tube and hepatoblasts by motif analysis. The promoters of 49 genes (including *Alb, C3, Apo,* and *Serpin* family members) highly expressed in the hepatoblasts had putative RXRA elements (Supplementary Data #8). The expression of these target genes increased in the liver primordium and peaked within the liver bud (Fig. 3f). Combined with IPA analysis, ALB and Serpins protein served as both the ligands and the targets in the LXR/RXR pathway, which implies a positive-feedback loop during liver specification (Fig. 3g). Besides RXRA, genes upregulated in hepatoblasts were found to be targets of HNF4A and PPARG, while the targets of SOX2 and TEAD1 were found in the downregulated genes (Supplementary Fig. 11). In conclusion, activation of the RXR complex signaling pathway and several TF genes, including *Lzts1,* are concomitant with the liver specification.

**Transient transcription factor gene expression during hepatoblast maturation into hepatocytes**. To study the dynamics of hepatoblast maturation into hepatocytes, we retrieved the *Alb*+ hepatoblast/hepatocytes at E11.5–E15.5 (Supplementary Fig. 12a, b). Combined with the liver primordium and liver bud cells at E9.5–E10.5, a trajectory of hepatic development was determined by Monocle[42] (Fig. 4a). Notably, there were no branches on the trajectory, and the predicted pseudotime of the trajectory agreed with the gestation day. During this timeline, we examined specific genes and gene sets defining the "hepatic score" (liver metabolic function genes including *Alb*), "stemness score" (stem markers including *Nanog*), and "proliferation score" (cell cycle genes, including *Mki67*). The metabolic function of hepatoblasts/hepatocytes increased while the cell pluripotency and the proliferation rate decreased during liver maturation (Fig. 4b, c). These results were also validated by the 720 cells generated by the MIRALCS method from the E11.5, E12.5, and E13.5 stages (Supplementary Fig. 12c).

With the Monocle analysis, we found 5869 genes dynamically regulated during hepatoblast maturation (*q* value < 0.01) (Supplementary Data #9). Interestingly, 85% (4974 genes) of these genes were downregulated, while only 15% (895 genes) were upregulated (Fig. 4d). The downregulated genes consisted of 12% TFs (582 genes), while the upregulated genes consisted of only 3% TFs (26 genes) (Fig. 4e). These results indicate that a large number of TFs play transient roles in liver specification and are decreased afterward. The upregulated genes were mostly related to the metabolic function of the liver (such as *Alb* and *Apoh*), while the downregulated genes were enriched in the cell cycle, RNA splicing, cell division, and translation (such as *Mdk* and *Set*) (Fig. 4d, f). Moreover, we found Ubiquitin B (*Ubb*) that regulated the protein ubiquitination process was downregulated during hepatoblasts maturation, perhaps to protect the metabolic enzymes and other proteins produced by the hepatocytes (Fig. 4g). The heat-shock response (HSR) pathway (genes including *Hsf1, Hsf2, Hsp70, Hsph1, Hspe1, Hspa8, Hsp90ab1,* and *Hsp90aa1*)

that control the protein folding process was also found to be downregulated (Fig. 4g).

## Discussion

Single-Cell RNA-Seq is a powerful tool in developmental biology. Super high-throughput single-cell RNA-Seq methods based on the droplet or split-pool technology have been widely used to describe the cell atlas of different organs or tissues. However, the low sensitivity of these methods affects the detection of the low expressed genes, including essential transcription factors during liver development. Here, we constructed a Foxa2[eGFP] mouse model and isolated and traced hepatic-related cells. We described high-quality full-length transcriptome data of primitive streak, gut endoderm, liver primordium, liver bud, and early fetal liver using SMART-seq2. The high-coverage transcriptome profiling (9378 detected genes/cell) allowed us to sensitively detect most of the transcriptional changes during liver development, especially the regulation of transcription factors.

We found that the transcriptional dynamics of embryonic liver development went through a "three-step" process. First, the E8.5 gut endoderm develops from the primitive streak; second, hepatoblasts are specified from the gut tube at E9.5; and finally, the hepatoblasts mature into hepatocytes (Fig. 5). These processes nicely agree with previous morphological studies[15] and reveal that the cells at each stage have markedly distinct transcriptional programs. We have described the key regulators and transcriptional dynamics during these processes.

To the best of our knowledge, this study is the first time to reveal the full-length transcriptome profile of liver primordium in E9.5, and described the dynamical gene expression of EHT process during liver specification. Six groups of genes (L1, L2, L3, G1, G2, G3) were fated to be switched on or off following the developmental axis GT–LP–LB–Liver (Fig. 3a, b). Different functions of each gene group provide potential clues on the sequence of significant events during liver development. *Lzts1* was identified and verified to be upregulated during liver development (Fig. 3d, e). Moreover, we found the EHT process to be associated with the activation of the LXR/RXR signaling pathway (Fig. 3f, g). A positive-feedback loop of the LXR/RXR signaling pathway could explain why the expression levels of *Alb* and *Serpin* family genes were increased over 1000-fold in a short time compared with the gut tube. RAR-deficient mice are not lethal but display abnormal liver development[43].

During the maturation of hepatoblasts into hepatocytes between E11.5 and E15.5, the transcriptome was relatively stable and changed gradually during this process (Fig. 4a–c), implying that the majority of liver specification occurs during E9.5–E10.5. In addition, the hepatic features increased, while stemness features and cell proliferation decreased. Interestingly, 85% of the dynamically expressed genes were downregulated, including many TFs (Fig. 4d, e). This suggests that many genes including TFs are necessary for organ specification and are turned off after the specification is completed.

Together with the liver primordium, we identified a group of gallbladder primordium cells. Interestingly, we found the gallbladder primordium expressed many hepatic marker genes, including *Hnf4a, Prox1, Foxa2, Dlk1,* and *Alb,* but did not express *Foxa1* (Fig. 2f). Furthermore, we also identified some potential markers and effectors for the gallbladder, expressed by genes such as *Crabp1, Rab38, Flt1, Slco5a1, Ptpn5, Vstm2b, Ntrk2, Ets1,* and *Ypel4*.

More efforts are needed to understand liver development better in the future. In vivo knockout experiments are required to explore the functions of specific TFs in our dataset during liver specification. The heterogeneity of gut endoderm at E8.5 can be elucidated better by recruiting more gut endodermal cells.

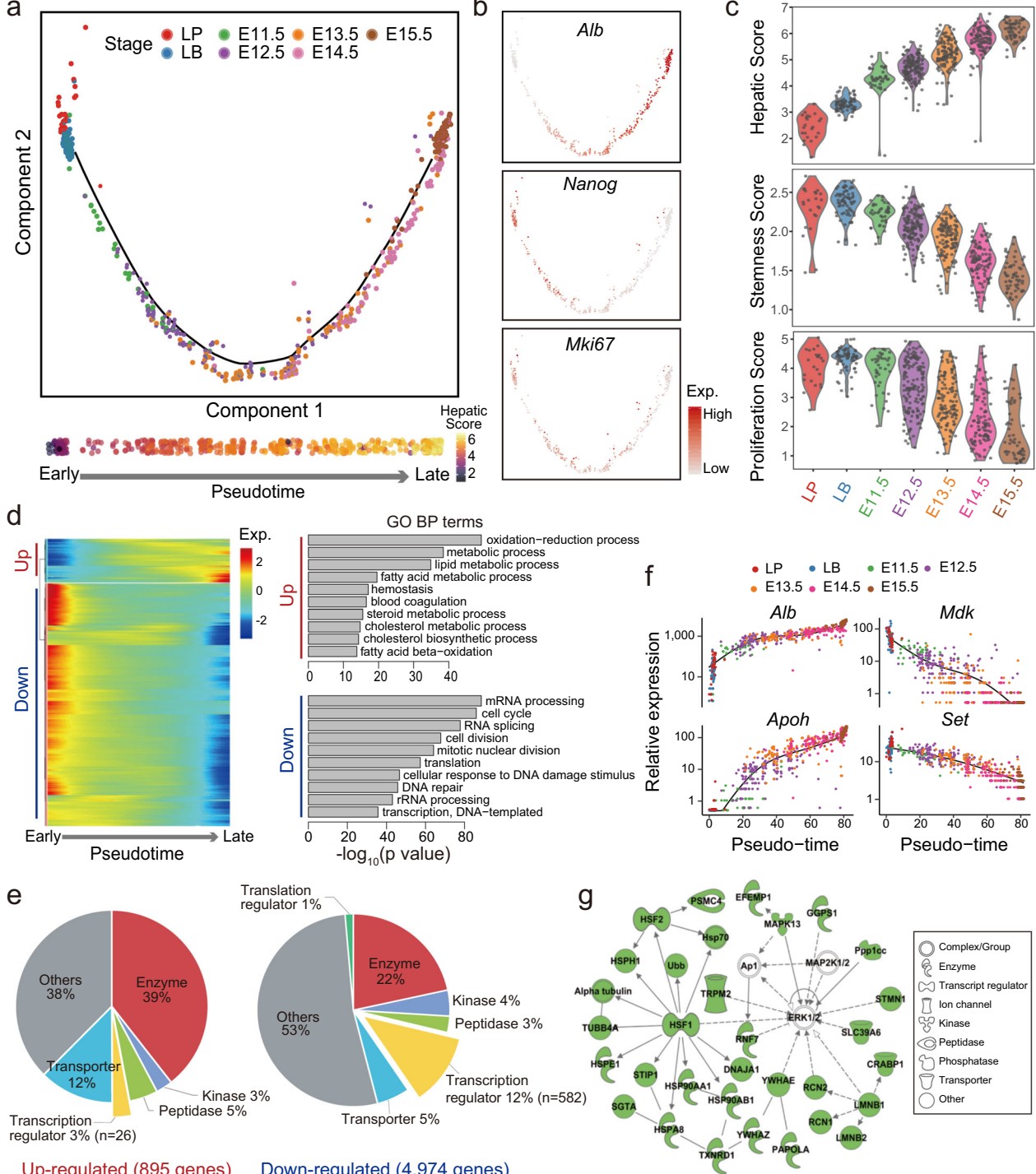

**Fig. 4 Dynamic gene expression during hepatoblast maturation into hepatocytes. a** A trajectory of hepatic development was determined during E9.5–E15.5 by Monocle analysis. No branches on the trajectory were found, and the predicted pseudotime of the trajectory agreed with the gestation day. **b** The expressions of *Alb, Nanog, Mki67* were shown in the hepatic trajectory. **c** The metabolic function of hepatoblasts/hepatocytes increased while the cell pluripotency and the proliferation rate decreased during liver development based on "hepatic score", "stemness score", and "proliferation score" from E9.5 to E15.5. **d** Dynamic regulation of gene expression during hepatoblast maturation and the respective functions of up/downregulated genes were identified. **e** The composition of upregulated or downregulated genes during E9.5–E15.5 were shown. **f** Genes for the metabolic function of the liver (such as *Alb* and *Apoh*) from E9.5 to E15.5 were upregulated, while the downregulated genes were enriched in cell cycles, RNA splicing, cell division, and translation (such as *Mdk* and *Set*). **g** The network of genes that were downregulated during the hepatoblasts maturation. Genes labeled by green were downregulated, while genes labeled by white were not detected.

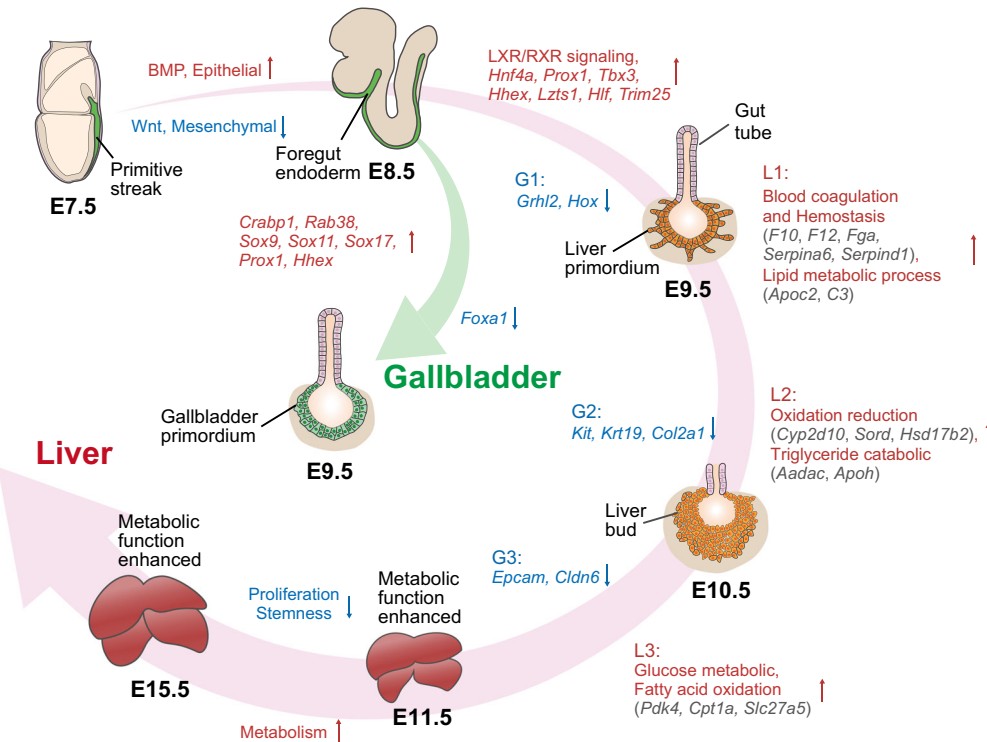

**Fig. 5 Dynamic transcriptome of liver and gallbladder development from the embryonic endoderm during E7.5–E15.5 by scRNA-Seq.** Genes/pathways/biological processes colored in red represent upregulation, while those colored in blue represent downregulation. Genes in gray are involved in related pathway/biological processes.

In summary, by tracing the *Foxa2* lineage with single-cell resolution, our study identified numerous key regulations critical for liver development, features that distinguish the gallbladder, and potential clues leading to therapeutically useful tissues for transplantation. Moreover, this Foxa2$^{eGFP}$ mouse model can be used to study the development of other endodermal organs, such as the lung and pancreas, which can provide insights into the mechanisms of endodermal organ development.

## Methods

**Targeting vector construction.** BAC clone RP23-469P2 containing mouse *Foxa2* locus was obtained from BACPAC as the template. Exon3 of *Foxa2* (Chr2: 147869253-147870662) was amplified by HiFi-PCR and inserted into pEYFPC1 by XhoI /EcoRI, then *eGFP*-coding region was amplified from pLEGFPC1 vector to fuse in-frame with Exon3 of *Foxa2* by linker CTA- GGA-ATT-CTA (Leu–Gly–Ile–Leu). Next, 3′UTR of *Foxa2* (Chr2: 147867876-147869250) was amplified from BAC clone and inserted after *Foxa2-eGFP* fusion by KpnI/BamHI on pEYFPC1 backbone. The whole region of Exon3 of *Foxa2*-eGFP-3′UTR of *Foxa2* was retrieved and inserted into pEASY-Flox vector by XbaI/SalI, which was flanked by the latter two LoxP sites. The first LoxP site and Neo cassette were removed, and restriction sites were introduced to facilitate downstream construction. In all, 4.5-kb *Foxa2* genomic region (Chr2: 147870690-147875176) before Exon3 was amplified from BAC and inserted into pEASY-flox by ClaI/BamHI. In all, 2.8-kb genomic region after 3′UTR of *Foxa2* (Chr2: 147865027-147867871) was amplified from BAC and inserted by HindIII/XhoI. The PGK (phosphoglycerate kinase) promoter-driven Neomycin expression cassette flanked by FRT sequence was inserted at the reverse direction by HindIII. The vector was partially verified by the sequencing of ligated regions. Genotyping was done by PCR using DNA extracted from tail tips from 3-week-old mice. The sequence of primer pairs used are provided:

Foxa2-eGFP-1: 5′-CTTTGGGGCCCAGAGGACTTGGTG-3′;
Foxa2-eGFP-2: 5′-GTATGTGTTCATGCCATTCATCCCCAGG-3′.
Foxa2-linker-eGFP:
5′-TATGAACTCATCCCTAGGAATTCTAGTGAGCAAGGGCGAG-3′.

**Mouse breeding and dissection/timed mating and embryos collection.** The Foxa2$^{eGFP}$ mice have been backcrossed to the C57BL/6 background for more than 20 generations and do not show any abnormal phenotypes. All animal procedures were complied with all relevant ethical regulations for animal testing and research and performed under the strict instruction by the Institutional Animal Care and

Use Committee (IACUC) approval (IACUC approval number R15-0831) of Comparative Medicine of National University of Singapore. All the mice used as the sample for Single-cell RNA-Seq were homozygous for Foxa2$^{eGFP}$. Two-to-three-month–year-old timed-pregnant Foxa2$^{eGFP}$ females were used for the sequencing.

**FACS.** Flow cytometry (BD FACSAria$^{TM}$ III cell sorter) was used to analyze and sort target cells. Cells collected from the wild-type mouse liver were used as control and set the eGFP negative gate on the cell sorter. Multiplets were excluded in our sorting. Single cells were sorted onto a glass slide and checked under a microscope before sorted into each well of 96-well plates. Each well of the plates was pre-loaded with 4 µl of cell lysis buffer. In all, 100–200 cells were sorted into 1–3 wells as bulk control, and 1 negative well with no cells was designed to evaluate the contamination of mRNA amplification.

**Single-cell mRNA amplification.** Message RNA from single cells was amplified using Smart-seq2[44] with modifications. Briefly, cells were lysed at 65 °C for 3 min and then subject to reverse transcription using an oligo(dT) primer and a locked nucleic acid (LNA)-containing template-switching oligonucleotide (TSO primer) (Exiqon). Full-length cDNAs were amplified by 20 cycles of PCR using KAPA HiFi DNA polymerase (KAPA biosystems) and IS primer. The products from randomly selected three wells and the negative well were run on the agarose gel. Only the plates that were successfully amplified without contamination were processed for library construction. All the products were purified with 0.8× Ampure XP beads (Beckman) and quantified with AccuBlue High Sensitivity dsDNA Quantitation Kit (Biotium). The amplified cDNA was assessed by agarose gel and qPCR before the library generation to ensure the sequencing quality (Supplementary Fig. 2).

Oligo(dT) primer: 5′-AAGCAGTGGTATCAACGCAGAGTACT (30) VN-3′
(V = G/A/C, N = G/A/C/T)
TSO primer: 5′-AAGCAGTGGTATCAACGCAGAGTACrGrG+G-3′
(rG=RNA Guanine, +G = LNA modified guanine)
IS primer: 5′-AAGCAGTGGTATCAACGCAGAGTAC-3′.

**Real-time PCR.** Real-time PCR was used to test the success of single-cell cDNA amplification. The cDNA products were subjected to RT–PCR with KAPA SYBR® FAST Universal 2X qPCR Master Mix40 (KK4600) using a 7300 or 7500 Real-Time PCR machine (Applied Biosystems). The mRNA expression of *Afp*, a hepatic marker, was detected and normalized against the internal housekeeping gene *Gapdh*.

*Afp* forward primer: 5′-GCTCACATCCACGAGGAGTGTT-3′
*Afp* reverse primer: 5′-CAGAAGCCTAGTTGGATCATGGG-3′

*Gapdh* forward primer: 5′-CATCACTGCCACCCAGAAGACTG-3′
*Gapdh* reverse primer: 5′-ATGCCAGTGAGCTTCCCGTTCAG-3′.

**Embryo fixation**. The whole embryo was fixed in 4% paraformaldehyde for 1–2 h at room temperature. After fixation, tissue was rinsed with PBS until fixative is completely removed. Tissue was dehydrated by using different series of ethanol, citrisolve, and paraffin: 50% ethanol for 10 min; 70% ethanol for 10 min; 80% ethanol for 10 min; 95% ethanol for 10 min; 100% ethanol for 10 min; 100% ethanol for 10 min; 100% ethanol for 10 min; 2:1 ethanol: citrisolve for 10–15 min; 1:1 ethanol: citrisolve for 10–15 min; 1:2 ethanol: citrisolve for 10–15 min; 100% citrisolve for 10–15 min; 100% citrisolve for 10–15 min; 100% citrisolve for 10–15 min; 2:1 citrisolve: paraffin for 30 min; 1:1 citrisolve: paraffin for 30 min; 1:2 citrisolve: paraffin for 30 min; 100% paraffin for 1–2 h; 100% paraffin for 1–2 h or overnight.

**Immunohistochemistry**. Tissues were either fixed overnight and embedded in paraffin. Sections were cut 4–5 mm thick. Paraffin sections were deparaffinized, dehydrated, and we performed antigen retrieval by steaming slides in sodium citrate buffer for 30 min. Sections were blocked in the blocking serum buffer (5% serum in PBS + 0.5% Triton X-100) for 30 min. Primary antibodies were diluted in blocking buffer and incubated on tissue sections overnight at 4 ℃. Slides were washed and incubated in secondary antibody in blocking buffer for 2 h at room temperature. Slides were washed and mounted using Fluormount-G and observed under a microscope. Antibody information: FOXA2 (Cell Signaling, #3143); DLK1 (Abcam, ab21682); HNF4A (LSBio, LS-C413074); LZTS1 (Bioss, bs-5705R).

**Library preparation and sequencing with BGISEQ-500**. All the cDNAs were converted into libraries and sequenced on the BGISEQ-500 sequencer. BGISEQ-500 is an industry-leading high-throughput sequencing solution, powered by combinatorial Probe-Anchor Synthesis (cPAS) and improved DNA Nanoballs (DNB) technology[45,46].

In all, 2 ng of the cDNA was fragmented using the Tn5 enzyme-adaptor compound. In total, 15 cycles of PCR were then carried out with barcoded primers compatible with the BGISEQ-500. The 300–500 bp DNA fragments were selected and purified. The fragments were then heat-denatured, and one of the single strands was circularized with DNA ligase to obtain a single-strand circular DNA library. The remaining single strand was digested with the exonuclease. The sequencing process was conducted according to the BGISEQ-500 protocol[46].

**Public dataset access**. Mouse (Mus musculus) reference genome (mm10)[47] was downloaded from http://genome.ucsc.edu/. The transcriptome reference annotation GTF file (Ensembl GRCm38)[48] was downloaded from http://www.ensembl.org/. The sequence of *eGFP* was inserted into the *Foxa2* in mm10 reference file based on the structure of the transgenic vector. The annotation of eGFP was also added to the GTF file.

**RNA-seq data processing**. The raw sequencing data were accessed by filtering reads with adapters/poly-A, N rate >0.05, or low-quality base rate >0.5 using SOAPnuke (v1.5.6)[49]. Clean reads were mapped to the reference by TopHat (v2.1.0)[50] using Bowtie (v0.12.9.0)[51] with parameters "--bowtie1 –p 4 -g 1 --solexa1.3-quals --fusion-search --fusion-min-dist 100000". The Bowtie index of mm10 was built on autosomes and chrX.

**Quantification of gene expression levels**. Gene expression levels were quantified as RPKM. Read counts were calculated by Rsubread[52] (v1.16.1), and RPKM values were calculated using edgeR[53] (v2.6.12) with the edited GTF file (GRCm38). Cells with mapping read <1 million or mapping rate <40% were discarded. Cells in 96-wells plates with detected gene number (RPKM > 1) >6000 were defined as qualified cells, and the threshold of gene number was set as 4000 for cells generated by MIRALCS. Cells with $RPKM_{Foxa2} \geq 1$ and $RPKM_{eGFP} \geq 1$ were defined as Foxa2 + cells.

**Sequencing data assessment**. We devised a pseudobulk by pooling all single cells from the same stage and compared their gene expression levels with that of bulk sample (Supplementary Fig. 5a). The correlations were extremely high (Pearson $r >$ 0.9), indicating that our method was accurate and sensitive. Additionally, the amplification bias was assessed by comparing the expression level of the *Foxa2* and *eGFP* since they were expected to be transcribed at the same time. We found a high correlation between *eGFP* and *Foxa2* ($r > 0.9$) in the Foxa2+ cells, indicating a low bias in amplification (Supplementary Fig. 4b, c). Moreover, we assessed the systematic error within a batch of repeat experiments using E12.5, E13.5, and E14.5 livers. The results showed high correlations between the two batches of data from the same developmental stage (Supplementary Fig. 5c). To assess the sequencing batch effect, we pooled the libraries from E9.5 and E10.5 and sequenced them on two separate sequencing chips (Supplementary Fig. 5d). Besides, libraries from E7.5 were also sequenced on two chips (Supplementary Fig. 5e). All these results showed a high correlation and low-sequencing batch effect.

**Cell clustering and marker genes identification**. In total, 922 cells from E7.5 to E15.5 were clustered by Seurat (v2.1.0)[54]. The read count matrix was firstly column-normalized and log-transformed. The high variable genes identified by "Find Variable Genes" function were used for PCA analysis. The appropriate PCs were selected for clustering with the specific resolution parameters. Then t-distributed stochastic neighbor embedding (t-SNE) was used with the same number of PCs to visualize the clustering results. To detect cluster-specific expressed genes (marker genes), clusters were compared pairwise using "Find All Markers" function. Genes with at least 0.5-fold difference (log-scale) and a detectable expression in more than 50% of cells in either population were identified as candidate marker genes.

The resolution parameter is important for cluster determination and should be adjusted according to cell number and latent cell types in dataset. The top PCs represent a robust compression of the dataset, and each of the top PCs essentially represent a "metafeature" that combines information across a correlated gene set. The "metafeature" in the dataset is affected by the cell number and cell heterogeneity. So, we ranked the PCs based on the percentage of variance explained by each one and chose the elbow point in the percentage of variance explained by successive PCs. For cells from E7.5 and E8.5, Foxa2+ cells were involved in the analysis, and the top 12 PCs were selected with a resolution parameter equal to 1. For cells from E9.5 and E10.5, Foxa2+ cells were used, and the top five PCs were selected with a resolution parameter equal to 0.8. For cells from E11.5 to E15.5, all cells were used, and the top eight PCs were selected with a resolution parameter equal to 0.7.

**Differential expression analysis**. We performed differential expression analysis using SCDE[55] (v1.99.1), which adopted a Bayesian approach fitting individual error models. We selected the genes whose value of the most likely fold expression difference was more than one (ce >1) as differentially expressed genes for subsequent analysis.

For cells from E9.5 to E10.5, the cells were separated into different groups based on the cell types identified by Seurat. We performed the Kruskal-Wallis test (KW test) and retained the genes with adjusted *q* value < 1e-4.

SCDE is computationally intensive and slow with an increasing number of cells, whereas the DESeq2 (embedded in RaceID) is generally faster. However, the dedicated single-cell methods (SCDE) performed best in terms of precision. Hence, we performed differential gene expression analysis for cells from E9.5 to E11.5 (*n* = 187) by RaceID. RaceID[56] was applied to analyze the dataset using default model parameters. The cluster-specific marker genes were selected using the "clustdiffgenes" of RaceID with *P* value equal to 0.05.

**Pesudotemporal analysis**. We construct pseudo-temporal analysis using Monocle2[57] (v2.6.4) in cells from E9.5 to E11.5. Only cells identified as "Liver primordium", "Liver bud", or "Hepatoblast" from E9.5 to E15.5 were included. The genes expressed in at least ten cells with RPKM > 1 were retained. The differentially expressed genes across different cell stages with *q* values <0.001 were used for pseudo-temporal analysis. After constructing the cell trajectories, differentially expressed genes along the pseudotime were detected using "differential Gene Test" function, and genes with *q* values <0.01 were retained.

**Ingenuity pathway analysis**. IPA was performed to interrogate the biological functions during the development. The differentially expressed genes between different cell types were uploaded into the IPA software for the core analysis and identified the canonical pathways, diseases and functions, upstream regulators, and gene networks.

**ITranscriptome analysis**. ITranscriptome[34] was used to identify the potential spatial locations of cells defined as a primitive streak (PS). The zipcode genes were downloaded from http://www.picb.ac.cn/hanlab/media/lmdseq/static/zipcodegenes.txt. The expression matrix of zipcode genes of PS cells was uploaded to perform Zipcode Mapping analysis.

**Motif analysis**. HOMER (Hypergeometric Optimization of Motif EnRichment, v4.8.3, http://homer.ucsd.edu/homer/motif/)[58] was applied for searching promoters of genes and motifs enriched in target gene promoters as well. The "findMotifs.pl" script was downloaded from the website to analyze the regulators involved in up/downregulation of differential expression genes. Cluster-specific expressed genes were used to search for motifs with length 8, 10, or 12 from −300 to +100 relative to the transcription start site (TSS).

**Gene ontology analysis and feature score quantification**. We performed gene ontology analysis using the DAVID Bioinformatics Resource v.6.8 (https://david.ncifcrf.gov/). The feature scores were calculated by the average expression (log2-transformed) of each feature gene set (Supplementary Data #4).

**Statistics and reproducibility**. Signature score comparisons between different cell types were performed using unpaired two-tailed Student's *t* test. Differential expression genes between two cell types were tested with individual cell error

models implanted in SCDE package. Genes with absolute value of adjusted Z score more than 1 were identified as Differential expression genes. All statistical analyses and presentation were performed using R. Samples from the hepatic regions from E12.5, E13.5, and E14.5 were repeated by SMART-seq2. In addition to single cells from 96-well plates, libraries from 720 cells from E11.5, E12.5, and E13.5 by MIRALCS (microwell full-length mRNA amplification and library construction system) were generated.

**Reporting summary**. Further information on research design is available in the Nature Research Reporting Summary linked to this article.

## Data availability
The raw and processed data generated in this study have been deposited into CNGB Sequence Archive[59] (CNSA: https://db.cngb.org/cnsa) of CNGBdb with accession number CNP0000236. Previously generated drop-based single-cell RNA-sequencing data for mouse embryos analyzed in this study can be downloaded from the NCBI Gene Expression Omnibus with accession number GSE119945. Another full-length single-cell RNA-seq for the development of mouse embryos hepatocyte was acquired from the NCBI GEO repository with accession number GSE90047. The plasmids in this study are deposited in GenBank (accession number MT936307). In addition, any relevant data upon request is available by contacting the corresponding author (Dr. Yong Hou).

## Code availability
The source code for single-cell bioinformatics analysis in our study is available at GitHub (https://github.com/CellOmics-Yu/Mus_liver_development).

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

## Acknowledgements

We thank Dr. Zakir Hossain for mouse embryo electroporation; Dr. Tang Fuchou for comments on the paper; Dr. Nicolas Plachta for the 3D-imaging of mouse embryo culture; Dr. Chengran Xu for demonstration of mouse manipulation; Dr. Eseng Lai, Robert H. Costa, and James E. Darnell Jr. for intellectual contributions to the Foxa2 study. The work at Xin-Yuan Fu's lab was supported by grants from the Singapore National Medical Research Council (R-713-000-181-511), the Ministry of Education (R-713-000-169-112), Cancer Science Institute of Singapore (CSI, Singapore) (to X.-Y.F. grant 713001010271), the Office of Deputy President (DPRT) of National University of Singapore, West China Hospital (Sichuan University, China) (No. 139170082), the President Fund of SUSTech (22/Y01226237) and funds from GenEros Biopharma. The work at BGI-Shenzhen was supported by grants from the National Natural Science Foundation of China (No. 81672593), the Natural Science Foundation of Guangdong Province, China (No. 2018A030313379) and Science, Technology and Innovation Commission of Shenzhen Municipality (Nos. GJHZ20180419190827179 and KQJSCX20170322143848413). The work was also supported by funding from the Institute for Regenerative Medicine at UPenn to K.S.Z.

## Author contributions

X.-Y.F., T.M., and X. Liu conceived the project. X. Liu and T.M. designed the mouse model. T.M. performed the acquisition and preparation of hepatic samples. Liqin Xu and T.M. performed scRNA-Seq and sequencing. Liqin Xu, T.M., Y.Z., and Z.Z. conducted the bioinformatics calculation. T.M. and Liqin Xu performed data analysis. T.M., C.H., and X. Lan conducted experimental verification. C.L., Yi Zhou, Y.S., Luang Xu, M.J., H. Z., X. Lin, Liang Wu, S.P., and S.L. contributed to the project execution. T.M. and Liqin Xu wrote the paper. S.B., M.D., N.R.D., K.S.Z., X.-Y.F., and Y.H. revised the paper for important intellectual content.

## Competing interests

The authors declare no competing interests.
