## [Peer Review File · Communications Biology]

Reviewers' comments:

Reviewer #1 (Remarks to the Author):

In this work Mu and colleagues use single-cell RNA-seq (scRNA-seq) to identify the developmental trajectory of embryonic liver from gut endoderm to hepatoblasts and characterize the transcriptome of the hepatic lineage.

They found both gut and liver features in the emerging liver primordium and dynamic gene expression changes during the epithelial-hepatic transition (EHT) at the stage of liver specification from E9.5-E11.5, with six groups of genes switched on or off in the EHT process. RXR signaling and transcription factors including *Lzts1* were identified as potential regulators of EHT. Additionally, the authors identified the transcriptional profile of the gallbladder primordium at E9.5.

Together the data provides a high-resolution map of gene expression changes during hepatic specification that should be useful for understanding the liver and gallbladder development.

Critique:

Overall this study is clearly described, technically well executed and does provide an important resource. However, the study is exclusively descriptive and no effort has been made to follow up predictions about for example RXR signaling or *Lzts1* as regulators of EHT. This is a weakness that should be addressed in a revised version.

Reviewer #2 (Remarks to the Author):

Mu and colleagues performed single-cell RNA sequencing on the *FOXA2* positive cells, endogenously labeled with eGFP in the transgenic mice. Authors have isolated these eGFP-positive cells from different stages of the embryonic mice to understand the liver and gallbladder organogenesis at the single-cell resolution. In addition to the scientific advancements, the manuscript provides an enormous amount of genomics data, and therefore, could serve as a great resource for the community. Moreover, authors have appropriately taken all the necessary quality check measurements to ensure that all the claims are made from the good quality cells.

In order to further improve the quality of the manuscript, I have a few specific comments:

1. To ensure the high quality of the cells, authors have used various filtering criteria, including the minimum expression, the minimal number of expressed genes per cell, etc. It will be informative for the readers if authors can graphically depict these parameters, and could clearly indicate the total number of cells after each filtering criteria, further segregated into distinct developmental time points. This new figure will give an overview of the sanity checks and the information about the number of contributing cell-types at each developmental time-point. So far all this information is in the text.

2. Authors have performed standard single-cell bioinformatics analysis, including DGE, pseudotemporal analysis, etc. Although the methods are explained well, but to really ensure the reproducibility of each figure panels, authors must provide the codes in GitHub, or at least for the reviewer's assessment.

3. Importantly, although the authors have shown a good correlation between EGFP and FOXA2 gene across all selected genes, still it does not address the point that the transgenic mice indeed recapitulate the endogenous expression of FOXA2 (eGFP-/FOXA2+ population?). Therefore, since this piece of information is vital for all the downstream claims, the authors must quantitatively address this issue. The authors can perform the whole-mount two-color immunohistochemistry/in situ hybridization with anti-eGFP and in situ hybridization (probe for endogenous FOXA2). The authors should quantitatively represent the findings.

4. Quantification of the co-labeling experiment (Figure 1D) is missing.

If these concerns are addressed, the manuscript is a good fit for CommsBio.

Point-by-Point Response to Reviewers' Comments:

Response to Reviewer 1

Characterizing the Emergence of Liver and Gallbladder from the Embryonic Endoderm through Single-Cell RNA-Seq

Tianhao Mu, Liqin Xu, Yu Zhong, Xinyu Liu, Zhikun Zhao, Chaoben Huang, Xiaofeng Lan, Chengchen Lufei, Yi Zhou, Yixun Su, Luang Xu, Miaomiao Jiang, Hongpo Zhou, Xinxin Lin, Liang Wu, Siqi Peng, Shiping Liu, Susanne Brix, Michael Dean, Norris R. Dunn, Kenneth S. Zaret, Xin-Yuan Fu & Yong Hou

In this work Mu and colleagues use single-cell RNA-seq (scRNA-seq) to identify the developmental trajectory of embryonic liver from gut endoderm to hepatoblasts and characterize the transcriptome of the hepatic lineage. They found both gut and liver features in the emerging liver primordium and dynamic gene expression changes during the epithelial-hepatic transition (EHT) at the stage of liver specification from E9.5-E11.5, with six groups of genes switched on or off in the EHT process. RXR signaling and transcription factors including *Lzts1* were identified as potential regulators of EHT. Additionally, the authors identified the transcriptional profile of the gallbladder primordium at E9.5.

Together the data provides a high-resolution map of gene expression changes during hepatic specification that should be useful for understanding the liver and gallbladder development.

Thanks for your comments. They are useful and provide an enhanced clarity to our readers! We appreciate your effort and time! In this rebuttal, the text in **blue** represents our response to your comments (in **black**) and the text (in **red**) denotes the changes made to the revised manuscript.

Critique:

Overall this study is clearly described, technically well executed and does provide an important resource. However, the study is exclusively descriptive and no effort has been made to follow up predictions about for example RXR signaling or *Lzts1* as regulators of EHT. This is a weakness that should be addressed in a revised version.

Thanks for the critique. We appreciate the reviewer's point that by including details about *Lzts1* and RXR in the abstract. It led to the suggestion that we had done extensive validation work on their function during liver specification; as such, we deleted that sentence from the abstract and modified the results to minimize implications about functions of the genes. On the other hand, we realized that our original abstract didn't sufficiently feature our exciting discovery of many new transcription factors that are induced at the time of liver specification, so we have added that point to the revised abstract. We feel that the revised paper provides better focus on the extensive new information that we have discovered, and thank the reviewer for raising their point.

Page 6 : "We found six groups of genes switched on or off in the EHT process, including diverse transcriptional regulators that had not been previously known to be expressed during EHT."

Then, in the text page 17, we tweaked the header of the relevant section a bit, to soften the claims.

Page 17: “Significant transcription factors and RXR complex signaling dynamics during liver specification”

For the predictions for LXR/RXR signaling, we did described some interpretations for the role of LXR/RXR during EHT in the previous manuscript (Page 18, cited below), but perhaps the explanation was too long and not sufficiently understandable. By Ingenuity pathway analysis (IPA), we found that LXR/RXR signaling was very highly upregulated based on the high expression of both upstream and downstream genes (Figure S10D). Interestingly, by Motif analysis, we found that the promoters of *Alb*, *C3*, *Apo* and *Serpin* had putative RXRA elements (Figure 3F). Together, *Alb* and *Serpin* served as both the ligands and the targets in LXR/RXR pathway. A positive-feedback loop of LXR/RXR pathway during EHT (Figure 3G) is implied and able to explain that *Alb* and *Serpin* family genes were increased over 1,000-fold in a short time in liver primordium compared with the gut tube.

Page 18:

“More importantly, we found the liver X receptors/retinoid X receptors (LXR/RXR) pathway was significantly up-regulated in the liver primordium compared with the gut tube, including *Alb*, *Ambp*, *ApoA1*, *ApoA2*, *ApoE*, *ApoF*, *ApoM*, *C3*, *Ttr*, *SerpinA1*, *SerpinF1*, *SerpinF2* and others (Figure S10D).

To validate the role of the LXR/RXR pathway, we analyzed the promoters of the differentially expressed genes between the gut tube and hepatoblasts by motif analysis. The promoters of 49 genes (including *Alb*, *C3*, *Apo* and *Serpin* family members) highly expressed in the hepatoblasts had putative RXRA elements (Table S8). The expression of these target genes increased in the liver primordium and peaked within the liver bud (Figure 3F). Combined with IPA analysis, *Alb* and *Serpin* family served as both the ligands and the targets in the LXR/RXR pathway, which implies a positive-feedback loop during liver specification (Figure 3G).”

F

G

As a transcription factor, *Lzts1* was barely not expressed in gut tube and started to be expressed in liver primordium by our finding, which was not reported previously. We validated the expression of *Lzts1* in hepatoblasts at E11.5 by immunofluorescence to support our finding (Figure 3E). We tried to explore more information related to *Lzts1* during EHT. Unfortunately, no strong evidence was found about the connection between other genes and *Lzts1* during EHT, either by pathway analysis, motif analysis or network analysis. This is might due to limited information about the function of *Lzts1*. Our data presents a preliminary discovery of expression of *Lzts1* in hepatoblasts, and more investigation such as knock-out experiments is needed to reveal the role of *Lzts1* during liver development in the future.

E

Characterizing the Emergence of Liver and Gallbladder from the Embryonic Endoderm through Single-Cell RNA-Seq

Tianhao Mu, Liqin Xu, Yu Zhong, Xinyu Liu, Zhikun Zhao, Chaoben Huang, Xiaofeng Lan, Chengchen Lufei, Yi Zhou, Yixun Su, Luang Xu, Miaomiao Jiang, Hongpo Zhou, Xinxin Lin, Liang Wu, Siqi Peng, Shiping Liu, Susanne Brix, Michael Dean, Norris R. Dunn, Kenneth S. Zaret, Xin-Yuan Fu & Yong Hou

Mu and colleagues performed single-cell RNA sequencing on the FOXA2 positive cells, endogenously labeled with eGFP in the transgenic mice. Authors have isolated these eGFP-positive cells from different stages of the embryonic mice to understand the liver and gallbladder organogenesis at the single-cell resolution. In addition to the scientific advancements, the manuscript provides an enormous amount of genomics data, and therefore, could serve as a great resource for the community. Moreover, authors have appropriately taken all the necessary quality check measurements to ensure that all the claims are made from the good quality cells.

Thanks for your comments. They are useful and provide an enhanced clarity to our readers! We appreciate your effort and time! In this rebuttal, the text in **blue** represents our response to your comments (in **black**) and the text (in **red**) denotes the changes made to the revised manuscript.

In order to further improve the quality of the manuscript, I have a few specific comments:

1. To ensure the high quality of the cells, authors have used various filtering criteria, including the minimum expression, the minimal number of expressed genes per cell, etc. It will be informative for the readers if authors can graphically depict these parameters, and could clearly indicate the total number of cells after each filtering criteria, further segregated into distinct developmental time points. This new figure will give an overview of the sanity checks and the information about the number of contributing cell-types at each developmental time-point. So far all this information is in the text.

Very good suggestion and absolutely! To make the manuscript more informative for the readers, in this revised manuscript, we have graphically depicted the parameters suggested by the reviewer 2, including minimal expressed gene number per cell, minimal mapped reads per cell, cell sub-clusters (Foxa2+/-) and cell number for each cluster at different developmental time points (Supplemental Fig. S7A of revised manuscript).

Figure S7. Mapping the endoderm development.

A, Graphical depiction of different parameters at each developmental time-point. The parameters, including minimal expressed gene number per cell, minimal mapped reads per cell, cell sub-clusters (*Foxa2*^{+/-}) and cell number for each cluster at different developmental time-points are shown.

2. Authors have performed standard single-cell bioinformatics analysis, including DGE, pseudotemporal analysis, etc. Although the methods are explained well, but to really ensure the reproducibility of each figure panels, authors must provide the codes in GitHub, or at least for the reviewer's assessment.

Thank for this constructive comment! We have provided the source codes for single-cell bioinformatics analysis in our study and they can be accessed on Github (https://github.com/CellOmics-Yu/Mus_liver_development) now.

Changes in "Code availability" (Supplementary Methods, Line 225):

The source codes for single-cell bioinformatics analysis in our study can be accessed on GitHub (https://github.com/CellOmics-Yu/Mus_liver_development).

3. Importantly, although the authors have shown a good correlation between EGFP and FOXA2 gene across all selected genes, still it does not address the point that the transgenic mice indeed recapitulate the endogenous expression of FOXA2 (eGFP-/FOXA2⁺ population?). Therefore, since this piece of information is vital for all the downstream claims, the authors must quantitatively address this issue. The authors can perform the whole-mount two-color immunohistochemistry/in situ hybridization with anti-eGFP and in situ hybridization (probe for endogenous FOXA2). The authors should quantitatively represent the findings.

That is a very good point. I understand the concern that there might be some single-cells which were eGFP-/FOXA2⁺. However, we think this possibility is quite low. As *eGFP* was inserted to the exon3 of *endogenous Foxa2* locus (Figure 1A), this transgenic mice are able to isolate single-cells with the endogenous expression of *Foxa2* by sorting eGFP fluorescence. These two genes were linked together, shared the same promoter of endogenous *Foxa2*, and should be co-expressed theoretically.

At the beginning, we did try to co-stain both FOXA2 and eGFP by immunofluorescence to show that the two proteins are co-expressed in hepatoblasts. We tried two eGFP antibody (Santa Cruz, sc-8334 and sc-9996), and unfortunately both antibodies failed to stain eGFP. One possible reason is that these two eGFP antibodies were both raised by full length GFP (amino acids 1-238) and N-terminal of eGFP was linked to C-terminal of FOXA2, which led to the failure of eGFP staining.

We have carefully calculated the correlation index between the expression of *Foxa2* and *eGFP* quantitatively (Pearson $r=0.95$) (Figure 1E). In Figure 1E, we can find that different single-cells have various expression of *Foxa2/eGFP*. However, in a particular single-cell, the expression of *Foxa2* is generally equal to the expression of *eGFP*. Moreover, we can conclude there is basically no *Foxa2+ /eGFP-* single-cells, which confirmed that *eGFP* and *Foxa2* were co-expressed. This way is more sensitive than in situ hybridization assay to quantify the expression level of *Foxa2* and *eGFP* to draw the conclusion that *eGFP* and *Foxa2* were co-expressed in single-cells.

E

4. Quantification of the co-labeling experiment (Figure 1D) is missing.

Thanks for your timely suggestion! We have done three replicates (three slides) of the co-labeling experiments of DLK1 and FOXA2. We also quantified the cell number of FOXA2+/DLK1+ cells, FOXA2+/DLK1- cells, and FOXA2-/DLK1+ cells by Venn plot (Figure S1E, below). The results quantitatively verified that FOXA2 and DLK1 were co-expressed.

S1E, Quantification of co-labeling immunofluorescence assay of FOXA2 and DLK1 of three replicates at E12.5 by Venn plot.

Page 9: Immunofluorescence assay showed that hepatoblasts expressed FOXA2 and DLK1 at E12.5 simultaneously in *Foxa2^{eGFP}* mice (Figure 1D, S1E).

If these concerns are addressed, the manuscript is a good fit for CommsBio.

REVIEWERS' COMMENTS:

Reviewer #1 (Remarks to the Author):

In the current revision the authors have addressed some of the reviewers' concerns, but failed to address others.

Major point 1: Reviewer 1's concerns about this being a purely descriptive study with no efforts to experimentally test the predictions they make based on the scRNA-seq analyses is still a concern.

Major point 2: Reviewer 2's concern about co-expression of Foxa2 and eGFP and request for Foxa2/eGFP co-staining has apparently been attempted using two Santa Cruz antibodies that didn't detect the fusion protein. The authors suggest that the fusion protein context is preventing the antibodies from detecting eGFP. Were positive controls using FL eGFP included? Given that there is a plethora of anti-eGFP antibodies it should be possible to perform this analysis, which is critical as Reviewer 2 points out. The authors could try: Chicken polyclonal anti-GFP (IF 1:1000) Abcam Cat#ab13970; RRID:AB_300798 or Rabbit polyclonal anti-GFP (IF 1:1000) Clontech (Takara Bio) Cat#632460; RRID:AB_2314544, which both detect eGFP when fused to the C-terminus of another protein.

Reviewer #2 (Remarks to the Author):

The authors have addressed all my concerns in the revised manuscript.

Response to Reviewers' Comments:

Response to Reviewer 1

Characterizing the Emergence of Liver and Gallbladder from the Embryonic Endoderm through Single-Cell RNA-Seq

Tianhao Mu, Liqin Xu, Yu Zhong, Xinyu Liu, Zhikun Zhao, Chaoben Huang, Xiaofeng Lan, Chengchen Lufei, Yi Zhou, Yixun Su, Luang Xu, Miaomiao Jiang, Hongpo Zhou, Xinxin Lin, Liang Wu, Siqi Peng, Shiping Liu, Susanne Brix, Michael Dean, Norris R. Dunn, Kenneth S. Zaret, Xin-Yuan Fu & Yong Hou

In the current revision the authors have addressed some of the reviewers' concerns, but failed to address others.

Major point 1: Reviewer 1's concerns about this being a purely descriptive study with no efforts to experimentally test the predictions they make based on the scRNA-seq analyses is still a concern.

Response 1: This paper is indeed mostly descriptive. However, we did validate some of our findings by wet experiments. For example, co-expression of FOXA2 and DLK1 was detected at E12.5 by immunofluorescence (Fig. 1d), which is consistent with the expression pattern by single-cell RNA-seq. The nascent hepatoblasts from liver primordium expressed both HNF4A and DLK1 at E9.5, which was also validated by immunofluorescence (Fig. 2h). To our best knowledge, this is the first evidence to show the expression of DLK1 and the potential to isolate the nascent hepatoblasts at E9.5. In addition to immunofluorescence assay, we also validated this conclusion by FACS. In Supplementary Fig. 8a, the nascent hepatoblasts at E9.5 were successfully sorted by FACS with DLK1-PE antibody, which proves the great value of our scRNA-seq data to the future studies. Moreover, as a new identified transcription factor during liver specification, the expression of LZTS1 was also validated (Fig. 3e), supporting our finding from scRNA-seq. Therefore, we humbly beg to differ with the point that our study is a purely descriptive study with no efforts to experimentally test the predictions. In our study, we validated some important findings during liver development to improve the confidence and significance of our study. Limited to huge amounts of information about scRNA-seq during liver development, it is very difficult to validate every single prediction from scRNA-seq data, at least under the current technology. Hope this could be understood.

Major point 2: Reviewer 2's concern about co-expression of Foxa2 and eGFP and request for Foxa2/eGFP co-staining has apparently been attempted using two Santa Cruz antibodies that didn't detect the fusion protein. The authors suggest that the fusion protein context is preventing the antibodies from detecting eGFP. Were positive controls using FL eGFP included? Given that there is a plethora of anti-eGFP antibodies it should be possible to perform this analysis, which is critical as Reviewer 2 points out. The authors could try: Chicken polyclonal anti-GFP (IF 1:1000) Abcam Cat#ab13970; RRID:AB_300798 or

Rabbit polyclonal anti-GFP (IF 1:1000) Clontech (Takara Bio) Cat#632460; RRID:AB_2314544, which both detect eGFP when fused to the C-terminus of another protein.

Response 2: The positive controls using both FL eGFP antibodies were included and both FL eGFP worked well. However, either these two FL eGFP antibodies were not able to detect FOXA2-eGFP fusion protein. Thanks very much for the recommendations of other eGFP antibodies! It will help a lot when using our Foxa2-eGFP mouse model in the future. We are planning to purchase these antibodies and test their abilities.

Reviewer #2 (Remarks to the Author):

The authors have addressed all my concerns in the revised manuscript.

Response: Thanks for the comments. We are so glad that the revised manuscript has addressed your concerns!